# META-LEARNING WITH TASK-ENVIRONMENT INTERACTION

## ABSTRACT

The goal of meta-learning is to learn a universal model from various meta-training tasks, enabling rapid adaptation to new tasks with minimal training. Currently, mainstream meta-learning algorithms randomly sample meta-training tasks from a task pool, and the meta-model treats these sampled tasks equally without discrimination, training on them as a whole. However, due to the limitations imposed by training computational power and time constraints, harmful tasks sampled from the imbalanced distribution can have a significant impact on the optimization of the meta-model. Therefore, this paper introduces a form of meta-learning called Task-Environment Interaction Meta-Learning(TIML), which is distinct from reinforcement learning with data preprocessing. In TIML, we create a Task Exploration and Exploitation Selector that assesses the interaction between the meta-learning model and the presently sampled task environment. It conducts training differently based on factors such as task difficulty, rewards, harmfulness levels, and others, thereby altering the current practice of uniformly handling multiple tasks. By doing so, we can rapidly enhance the generalization and convergence of meta-learning parameters for unknown tasks. Experimental results demonstrate that the proposed TIML method achieves improvements in model performance while maintaining the same training time complexity. It exhibits faster convergence, greater stability, and can be flexibly combined with other models, showcasing its robust simplicity and universality.

## 1 INTRODUCTION

True artificial intelligence should possess the ability to learn how to learn, enabling rapid proficiency in few-shot or even zero-shot learning scenarios. For machines, the strategy is to commence with prior knowledge and transfer experiences from analogous tasks to novel ones. Meta-learning (Schmidhuber, 1987; Thrun & Pratt, 2012) stands as a framework that embodies the entire learning process, extracting common knowledge from tasks within the same distribution.

Presently, meta-learning algorithms can be categorized into three primary types: metric-based, network-based, and optimization-based algorithms. Metric-based techniques encode prior knowledge into an embedding space where similar (different) classes are brought closer (pushed farther) apart (Liu et al., 2018; Koch et al., 2015; Snell et al., 2017; Sung et al., 2018; Vinyals et al., 2016; Oreshkin et al., 2018). Conversely, network-based methods, often seen as black-box methods, employ networks or external memory to directly generate weights (Munkhdalai & Yu, 2017; Munkhdalai et al., 2018; Santoro et al., 2016), weight updates (Andrychowicz et al., 2016; Hochreiter et al., 2001; Ravi & Larochelle, 2017; Antoniou et al., 2018; Nichol et al., 2018; Finn et al., 2017b; Rajeswaran et al., 2019), or predictions (Mishra et al., 2017; Santoro et al., 2016). Meanwhile, optimization-based methods utilize a two-level optimization approach to comprehend the learning process. This encompasses initialization and weight updates, tailored for adapting to new tasks with minimal examples (Antoniou et al., 2018; Baik et al., 2020a;b; Jiang et al., 2019; Finn et al., 2017b; Wu et al., 2023). The outer-level optimization fosters generalization, while the inner-level optimization facilitates adaptation to new tasks, working in tandem to achieve meta-learning.

Take, for example, Model Agnostic Meta-Learning (MAML) Finn et al. (2017b), which learns an effective initialization that enhances few-shot learning's generalization prowess. The model rapidly adapts to new tasks through minimal gradient updates. Following this trajectory, recent research

endeavors have concentrated on refining initializations. For instance, learning adaptive hyperparameters (Baik et al., 2020a; Li et al., 2017) has yielded a novel weight update rule that significantly enhances the rapid adaptation process. Incorporating adaptive loss functions Baik et al. (2021) into various tasks results in more precise gradient updates, bolstering generalization for future tasks. The practice of conducting large-scale Shin et al. (2021) inner-loop meta update bias can enhance model adaptability across a broader spectrum of tasks. Some approaches even involve adjusting initial parameters through data augmentation (Rajendran et al., 2020; Ni et al., 2021) or altering the random sampling methodology Liu et al. (2020). In order to learn the meta-model during meta-training, most existing meta-learning methods employ a uniform probability random sampling of meta-training tasks. The underlying assumption behind this uniform sampling is that all tasks are equally important, but in reality, this is often not the case. The number of meta-training tasks may be limited, leading to an uneven distribution across different task clusters. This has given rise to various meta-learning methods (Kaddour et al., 2020; Yao et al., 2021; Liu et al., 2020) that address the sampling process, making judgments about unsampled tasks and selecting tasks that are more beneficial for training. This approach can be seen as a form of data preprocessing. While this approach may appear promising in terms of results, our goal is to strive for machine learning that resembles the human brain. In the process of natural selection, we cannot choose whether tomorrow will be sunny or rainy; attempting to change the environment we will encounter tomorrow goes against the principles of evolution. So, what can we do? When facing unknown environments, we can choose how to adapt rather than just taking a few steps in every direction, hoping to cover all possibilities. Since we lack the ability to change the environment, our best course of action is to adapt to it. Therefore, we have developed a task exploration and exploitation selector. When confronted with randomly sampled unknown tasks, we systematically explore each task to obtain initial feedback. Based on this feedback, we calculate future rewards and select tasks with greater potential for further training. We employ a step-by-step strategy, assessing whether to switch tasks after completing each one. In each task selection step, we introduce randomization to our decision-making process, adding an element of exploration to prevent falling into local traps.

## 2 RELATED WORK

The objective of few-shot learning is to tackle scenarios where only a limited number of samples are available for each task. The ultimate aim is to acquire the ability to generalize to unseen examples based on these constrained samples. To achieve this, the field of meta-learning places its focus on rapidly adapting to novel tasks through the accumulation of knowledge from a diverse array of similar tasks. Pioneering works in this domain encompass (Snell et al., 2017; Vinyals et al., 2016; Munkhdalai & Yu, 2017; Andrychowicz et al., 2016; Mishra et al., 2017; Santoro et al., 2016; Antoniou et al., 2018; Finn et al., 2017a; Edwards & Storkey, 2016; Finn et al., 2018; Lee et al., 2019; Wang et al., 2020; Zhou et al., 2021; Ravichandran et al., 2019; Tokmakov et al., 2019; Zhang et al., 2019; Bengio et al., 2013), among others.

One of the most widely acknowledged algorithms for acquiring an effective initialization is MAML (Finn et al., 2017b). Due to its straightforward nature and model-agnostic attributes, it has found extensive applications across various domains. The popularity of MAML has spurred the development of a series of variant algorithms rooted in MAML (Snell et al., 2017; Antoniou et al., 2018; Baik et al., 2020a;b; Jiang et al., 2019; Baik et al., 2021; Shin et al., 2021; Ni et al., 2021; Rajeswaran et al., 2019; Flennerhag et al., 2019; Grant et al., 2018; Jamal & Qi, 2019; Park & Oliva, 2019; Rusu et al., 2018; Triantafillou et al., 2019; Vuorio et al., 2019; Aimen et al., 2023). These algorithms aim to address known shortcomings of MAML, such as meta-level overfitting.

However, despite the success of these methods, they seem to overlook a crucial aspect: task environment interaction training. Many contemporary two-tier meta-learning algorithms lack explicit task interaction operations, and they train all tasks uniformly. There are also some indirect methods for handling tasks, such as adaptive task sampling algorithms Liu et al. (2020), which consider random task sampling to be suboptimal, and they achieve some results by greedy sampling of data. However, the time complexity of greedy sampling can significantly increase, and it may not adapt well to situations where the data environment cannot be changed. There are also data augmentation methods Ni et al. (2021) that can increase the amount of data for few-shot learning. Previous task handling methods can be summarized as preprocessing samples and then passing them to the model to learn better tasks. However, the goal of meta-learning is for any model to learn quickly under few-

shot conditions, and increasing data preprocessing methods can weaken the model's generalization, transferability, and handling of unknown situations.

Therefore, we propose a Task Environment Interaction Meta-Learning algorithm. We accomplish this by employing random task sampling to simulate natural random environmental occurrences. We have constructed a task environment interaction mechanism that selectively trains on randomly appearing task environments and includes a certain probability of conducting random exploration training. This approach achieves outstanding performance without altering the existing meta-model's structure and time complexity. It not only maintains its simplicity but can also be combined with other algorithms. Most importantly, it closely aligns with natural evolutionary scenarios where training for unknown tasks occurs randomly rather than in large-scale combinations.

## 3 APPROACH

### 3.1 PRELIMINARIES AND PROBLEM DEFINITION

Our objective is to acquire shared initial parameters $\varphi$ via a two-tier meta-learning model, which, after adapting to specific task learning, can offer promising starting points for various tasks. Leveraging existing knowledge, we can achieve favorable outcomes by taking a few steps of gradient descent. In the context of meta-learning, there exists a collection of tasks $\mathcal{T}_i$, where each task $\mathcal{T}_i$ is drawn from a task distribution $p(\mathcal{T})$. Each $\mathcal{T}_i$ comprises two separate datasets: a support set $\mathcal{D}_i^S$ and a query set $\mathcal{D}_i^Q$. Each task contains a set of inputs $x$ and outputs $y$: $\mathcal{D}_i^S = x_i^s, y_{i_{s=1}}^{sK}, \mathcal{D}_i^Q = x_i^q, y_{i_{q=1}}^{qM}$. We initialize the meta-parameter $\varphi$ in the outer loop of meta-learning and utilize it as the starting point for training in the inner loop of meta-learning. We randomly select multiple tasks $\mathcal{T}_i$ from the task distribution $p(\mathcal{T})$ and employ the support set $\mathcal{D}_i^S$ to train and update $\varphi_i$ for each task $\mathcal{T}_i$. Within each task $\mathcal{T}_i$, we employ the support set $\mathcal{D}_i^S$ to update the parameters $\varphi_i$ and employ the query set $\mathcal{D}i^Q$ to assess the performance of the updated parameters on that task, thereby obtaining the loss $\mathcal{L}_{\mathcal{T}_i}^{\mathcal{D}_i^Q}$. Following several rounds of training, we use the losses $\mathcal{L}_{\mathcal{T}_i}$ from each task to adapt the meta-parameters $\varphi$, aiming to enhance the model's generalization capabilities.

The MAML algorithm, as a well-recognized dual-level meta-learning model Finn et al. (2017a), aims to incorporate prior knowledge from sampled tasks into the neural network parameters $\theta$. These parameters serve as shared initialization weights, enabling rapid adaptation to new tasks. The network commences its learning process with the initialized weights $\theta$ and progressively adjusts them to cater to each task $\mathcal{T}i$ over a specified number of inner loop iterations. The network's parameters for each task are denoted as $\theta i, j$, where $j$ indicates the current time step. The update equation for $\theta_{i,j}$ is as follows:

$$\theta_{i,j+1} = \theta_{i,j} - \alpha \nabla_{\boldsymbol{\theta}} \mathcal{L}_{\mathcal{T}_i}^{\mathcal{D}_i^S} \left( f_{\theta_{i,j}} \right) \tag{1}$$

At the onset of each task, $\theta_{i,0} = \theta$. Following $S$ rounds of inner-loop updates, each task acquires a network weight value $\theta_i' = \theta_{i,\delta}$ that aligns with its specific task requirements. To assess the model's generalization and gather feedback regarding the initial parameters for the current task, a query set $\mathcal{D}_i^Q$ is extracted from $\mathcal{T}_i$ and employed to compute the updated initialization weight $\theta$, reinforcing generalization across all tasks.

$$\theta \leftarrow \theta - \beta \nabla_{\theta} \sum_{\mathcal{T}_i} \mathcal{L}_{\mathcal{T}_i}^{\mathcal{D}_i^Q} \left( f_{\theta_i'} \right) \tag{2}$$

### 3.2 META-LEARNING WITH TASK-ENVIRONMENT INTERACTION(TIML)

In the context of two-tier meta-learning, the outer loop is responsible for the meta-parameter's generalization ability, while the inner loop individually trains, validates, and provides feedback to the outer loop for each task $\mathcal{T}_i \sim p(\mathcal{T})$. In previous methods, the training approach and frequency for each inner loop task were the same. However, due to the inherent differences among tasks, this uniform training was not ideal. Many algorithms have started to improve task sampling by grouping tasks with low differences together for uniform training, achieving promising results (Yao et al., 2021; Liu et al., 2020; Wang et al., 2021). Nevertheless, this approach assumes that all samples need to be traversed to provide feedback, which may not align with real-world scenarios. In the evolution

of future algorithms, tasks won't be pre-prepared en masse, waiting for our algorithms to learn. Instead, they will emerge over time. Therefore, random sampling is more in line with the principles of natural evolution. Under the premise of random sampling, to address the limitations of uniform training, we design a task environment interaction mechanism. Each task is treated as an environment, akin to the natural world's environment selection. When multiple tasks appear simultaneously, in order to change the status quo of uniform training, we take the following steps: (1)Initially, we train each task for a brief period to gain a preliminary understanding of the environment and obtain feedback. (2)Based on the difficulty of the environment and our future expectations, we select one task to dive deeper into training to promptly acquire new feedback. We repeat this process to select tasks for in-depth training. (3)To avoid the greediness of task selection based solely on computational results, we introduce a normal random selection mechanism to enhance generalization. (4)Finally, we use the feedback from the in-depth training of tasks to update the meta-parameters in the outer loop.

This approach helps adapt to the ever-changing task landscape in a more natural and effective manner while improving the overall meta-learning process. The whole optimization algorithm of TIML is illustrated in Algorithm 1.

---

**Algorithm 1** Meta-Learning with Task-Environment Interaction(TIML).

---

**Require:** Task distribution $p(\mathcal{T})$, step size hyperparameters $\alpha, \beta$
1: Randomly initialize meta $\theta$;
2: **while** not done **do**
3:      Sample batch of tasks $\mathcal{T}_i \sim p(\mathcal{T})$;
4:      **for** all $\mathcal{T}_i$ **do**
5:          Initialize $\theta_{i,0} = \theta$
6:          Compute $I_i = \left( \theta_i, \mathcal{D}_i^S, \mathcal{D}_i^Q, \mathcal{L}_{\mathcal{T}_i}^{\mathcal{D}_i^Q}, N_i \right)$
7:      **end for**
8:      **for** level selection step $t := 1$ to $\text{len}(\mathcal{T}_i)$ **do**
9:          Compute $A_t = \arg\max_i (R_i(t))$ or $A_t = \text{random}(1, \text{len}(\mathcal{T}_i))$ ( random $< \varepsilon$)
10:          **for** inner-loop step $j := 2$ to S **do**
11:              Evaluate $\nabla_\theta \mathcal{L}_{\mathcal{T}_{A_t}}^{\mathcal{D}_{A_t}^s} \left( f_{\theta_{A_t, j-1}} \right)$ with respect to $K$ examples
12:              Compute adapted parameters with gradient descent:
13:              $\theta_{A_t, j} = \theta_{A_t, j-1} - \alpha \nabla_\theta \mathcal{L}_{\mathcal{T}_{A_t}}^{\mathcal{D}_{A_t}^s} \left( f_{\theta_{A_t, j-1}} \right)$
14:              Evaluate $\mathcal{L}_{\mathcal{T}_{A_t}}^{\mathcal{D}_{A_t}^Q} \left( f_{\theta_{A_t, j}} \right)$ with respect to $K$ examples
15:          **end for**
16:          Update the average reward $R_{A_t}(t)$ and the number of times selected $N_{A_t}$
17:          $N_{A_t}(t) = N_{A_t}(t-1) + 1, R_{A_t}(t) = R_{A_t}(t-1) + \frac{\mathcal{L}_{\mathcal{T}_{A_t}}^{\mathcal{D}_{A_t}^Q}(t) - R_{A_t}(t-1)}{N_{A_t}(t)}$
18:      **end for**
19:      Perform gradient descent to update meta-weight:
20:      $\theta \leftarrow \theta - \beta \nabla_\theta \sum_{\mathcal{T}_{A_t}} \mathcal{L}_{\mathcal{T}_{A_t}}^{\mathcal{D}_{A_t}^Q} \left( f_{\theta_{A_t}} \right)$
21: **end while**

---

After randomly sampling tasks, initial information gathering is conducted for each task. We define task environment information as $\mathcal{I}_i$, which includes task parameters $\theta_i$, support set $\mathcal{D}_i^S$, query set $\mathcal{D}_i^Q$, loss $\mathcal{L}_{\mathcal{T}_i}^{\mathcal{D}_i^Q}$, and the number of in-depth learning iterations $\mathcal{N}_i$. The formula is as follows:

$$I_i = \left( \theta_i, \mathcal{D}_i^S, \mathcal{D}_i^Q, \mathcal{L}_{\mathcal{T}_i}^{\mathcal{D}_i^Q}, N_i \right) \tag{3}$$

Next, the task environment information is interacted with the meta-model to provide feedback. Task information $\mathcal{I}_i$ is passed into the meta-model to calculate the future utility $\mathcal{R}_i$ for each task in the $t$-th round of selection:

$$R_i(t) = R_i(t-1) + \frac{\mathcal{L}_{\mathcal{T}_i}^{\mathcal{D}_i^Q}(t) - R_i(t-1)}{N_i(t)} \tag{4}$$

After obtaining the returns for the $t$-th round, we employ a greedy approach to directly select the task $\mathcal{A}_t$ with the highest returns for in-depth learning and record its number of in-depth learning iterations $\mathcal{N}_i$:

$$A_t = \arg\max_i \left(R_i(t)\right) \tag{5}$$

$$N_i(t) = N_i(t-1) + 1 \tag{6}$$

Due to the risk of falling into local optima with a greedy approach, we introduce a random parameter, denoted as "random" for task exploration, as shown in Figure 1 and Algorithm 1. When "random.rand" is less than the exploration factor $\varepsilon$, we randomly select a task for in-depth learning. This approach effectively prevents overfitting while providing valuable support for the generalization of outer meta-parameters.

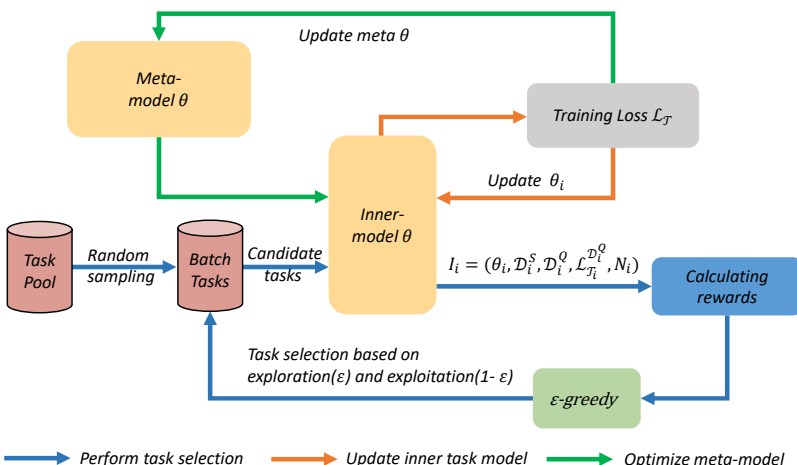

Figure 1: Illustration of TIML: (1) TIML interacts with task environment information for exploration or exploitation (blue arrows). (2) TIML engages in in-depth learning after task selection (orange arrows). (3) Task information from in-depth learning is used to update the meta-model (green arrows).

Here, we will also modify the original outer loop update mechanism. The proposed task-environment interaction mechanism, as shown in Figure 2, retains the information for each in-depth training task, denoted as $\mathcal{L}_{\mathcal{T}_{A_t}}^{\mathcal{D}_{A_t}^Q}\left(f_{\theta'_{A_t}}\right)$, during inner loop training. The tasks subjected to in-depth learning will be used for updating the meta-parameters.

$$\theta_{\text{meta}} \leftarrow \theta_{\text{meta}} - \beta \nabla_\theta \sum_{\mathcal{T}_{A_t}} \mathcal{L}_{\mathcal{T}_{A_t}}^{\mathcal{D}_{A_t}^Q}\left(f_{\theta'_{A_t}}\right) \tag{7}$$

## 4 EXPERIMENTS

In this section, we will demonstrate the effectiveness of Task Environment Interaction Meta-Learning in few-shot learning. We initiate the process with random task sampling, and after training, the meta-parameters are positioned closer to an initial state that better aligns with future tasks. By dynamically selecting tasks for training based on environmental interactions, we reduce the impact

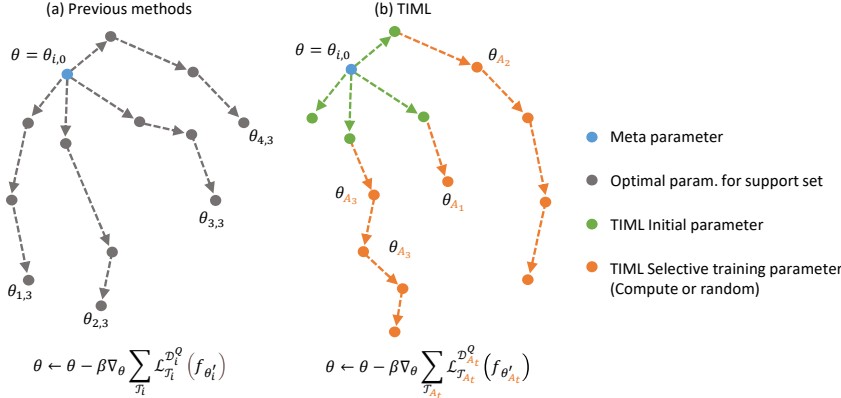

Figure 2: Inner Loop Optimization Overview of TIML for Few-shot Learning. (a) Previous methods uniformly train and learn each task $\mathcal{T}_i \sim p(\mathcal{T})$, updating meta-parameters based on the loss from all tasks $\mathcal{T}_i$. Due to compromises among tasks, meta-parameters remain in a conflicting state. (b) TIML dynamically updates the selection of tasks for in-depth learning in real-time through interaction between the meta-model and task environment information. It updates meta-parameters with the loss from the chosen tasks after training, breaking the previous task compromises and bringing meta-parameters closer to our desired position.

of harmful tasks on the meta-parameters, enabling us to reach optimization more swiftly and leading to smoother data fluctuations. When combined with other meta-learning algorithms, it enhances model accuracy from different perspectives. The experiments underscore the crucial role of task environment interaction training in the process of learning how to learn.

### 4.1 DATA SET

In the context of few-shot classification tasks, this article utilizes two widely employed datasets that have been prominent in recent meta-learning research: miniImagenet Simon et al. (2020) and tieredImagenet Finn et al. (2018). MiniImagenet is well-suited for the design and evaluation of few-shot learning and meta-learning model algorithms. Conversely, tieredImagenet is employed to assess the performance of meta-learning algorithms within more intricate and demanding environments.

MiniImagenet is created by randomly selecting 100 classes from the extensive ImageNet dataset. Subsequently, it is divided into three distinct subsets: a training set, a validation set, and a test set, with each subset containing images sized at 84x84 pixels. From these 100 classes, 64 are randomly designated for meta-training, 16 for meta-validation, and the remaining 20 for meta-testing. The dataset exhibits several key characteristics, making it a prevalent choice in the realm of meta-learning algorithms: (1)The dataset is relatively compact, enabling swift model training. (2)The classes within the dataset bear some resemblance to real-world scenarios, promoting applicability. (3)The dataset features a limited number of images per class, aligning perfectly with the few-shot learning paradigm and empowering the model to glean knowledge from scant examples.

TieredImagenet, while also derived from the ImageNet dataset, is structured differently based on supercategories. Comprising 34 supercategories in total, 20 supercategories are allocated for meta-training, 6 for meta-validation, and 8 for meta-testing. Each supercategory encompasses between 10 to 30 individual classes, leading to a grand total of 608 classes. This dataset presents a more formidable and intricate challenge, thereby offering a superior means of assessing the robustness and generalization capabilities of meta-learning algorithms.

In the context of cross-domain few-shot classification, to ascertain the model's effectiveness in adapting to novel tasks, we maintain the miniImagenet dataset as the training set and employ the test set of CUB-200-2011, commonly referred to as CUB[36], for evaluation purposes. CUB comprises 200 distinct bird species classes, similar in nature to miniImagenet, and serves as a widely adopted dataset for image classification. However, miniImagenet and CUB diverge in various as-

pects such as the number of categories, image size, and resolution. Notably, miniImagenet exhibits relatively low similarity between its classes, whereas CUB features more pronounced similarities. Consequently, the transfer from miniImagenet to CUB poses a formidable challenge within the realm of meta-learning.

## 4.2 IMPLEMENTATION DETAILS

In our experiments on datasets derived from ImageNet, we employ two distinct network architectures as the backbone feature extraction network, denoted as $f(\theta)$: a 4-layer CNN (referred to as 4-CONV) and the ResNet12 architecture. The architectural configurations for 4-CONV and ResNet12 closely adhere to the experimental settings established in prior studies (Finn et al., 2017a; Li et al., 2017; Nichol et al., 2018; Simon et al., 2020; Finn et al., 2018).

During the meta-training phase, our training regimen spans 100 epochs, each consisting of 500 iterations. For 5-shot and 1-shot scenarios, we utilize batch sizes of 2 and 4, respectively. Within each iteration, we undertake N-way classification by randomly sampling N classes. Subsequently, we further sample k labeled examples, denoted as $\mathcal{D}_i^S$, for the purpose of training, and an additional set of 15 examples, represented as $\mathcal{D}_i^Q$, for testing.

It is important to note that all experiments conducted on the original datasets adhere to the provided settings and hyperparameters established in their respective source code Mishra et al. (2017).

## 4.3 EXPERIMENTAL RESULTS

### 4.3.1 FEW-SHOT CLASSIFICATION

We conducted an evaluation of our TIML method using two network architectures: 4-CONV and ResNet12. In our experiments, we considered classification scenarios with 5 classes (5-way) and two different sample quantities: 5 samples per class (5-shot) and 1 sample per class (1-shot). We compared the performance of our method with various other meta-learning approaches on the miniImageNet and tieredImageNet datasets. The results, as summarized in Table 1, clearly demonstrate that TIML not only significantly enhances the generalization capabilities compared to MAML Finn et al. (2017b) but can also be effectively combined with other MAML variants like MAML++ Antoniou et al. (2018) and ALFA Baik et al. (2020a) to achieve even better performance. MAML++ focuses on learning fixed steps and per-layer inner loop learning rates, while ALFA adapts inner loop learning rates and regularization terms for each task. These methods contribute to the improvement of model performance from different angles.

TIML aims to design more rational training strategies for meta-learning models by engaging with the task environment, with the goal of rapidly advancing the model's evolution and development. In comparison to other methods, TIML can complement them, continuously refining the methodologies of meta-learning. The primary contribution of TIML lies in its ability to interact with randomly generated task environments in real-time, without the need for extensive data preprocessing. Prior to each training iteration, TIML selects tasks and then delves into training, simultaneously altering the update approach for meta-parameters to make the evolution of model parameters more realistic and intelligent.Our improvements go beyond enhancing the final model's generalization performance; more importantly, they enhance the model's intrinsic learning ability. TIML's approach no longer relies on extensive data preprocessing but instead changes the training methods and focus. It autonomously selects tasks that are beneficial for model training. Furthermore, to mitigate the risk of overfitting, a random selection mechanism is introduced. This represents a significant advancement in the field of artificial intelligence. We are not attempting to change the environment (data preprocessing) but rather striving to adapt to the environment through TIML.

In Table 1, a vertical comparison reveals the differences in performance between TIML and other meta-learning algorithms under the two network architectures, 4-CONV and ResNet12. It's evident that the improvement is more significant under the ResNet12 architecture compared to 4-CONV. This suggests that in more complex network structures, such as ResNet12, the TIML approach becomes increasingly important and exerts a greater impact on model generalization. Similarly, in the horizontal comparison, the improvement in the 5-shot setting is noticeably more pronounced compared to the 1-shot setting. This indicates that as the number of samples increases, the con-

flicts between tasks also escalate, emphasizing the critical importance of selecting appropriate tasks. Throughout the experimental results, we employed the variance of the 95% confidence interval. Under our task-environment interactive meta-learning approach, the variance consistently remained lower than that of other meta-learning methods. We enhance the algorithm's stability, generalization ability, reliability, and efficiency. These improvements are of paramount significance and render the algorithm more practical and reliable, making it suitable for a broader range of application scenarios.

Table 1: Test accuracy on 5-way classification for miniImageNet and tieredImageNet

| Model | Backbone | miniImagenet | | tieredImageNet | |
|---|---|---|---|---|---|
| | | 1-shot | 5-shot | 1-shot | 5-shot |
| MAMLFinn et al. (2017b) | 4-CONV | 48.7 ± 1.75% | 63.11± 0.91% | 49.06 ± 0.91% | 67.48 ± 0.47% |
| MAML + TIML(ours) | 4-CONV | **48.9 ± 0.49%** | **65.10 ± 0.47%** | **49.85 ± 0.48%** | **70.18 ± 0.35%** |
| MAML++Antoniou et al. (2018) | 4-CONV | 52.15 ± 0.26% | 68.32 ± 0.44% | —— | —— |
| MAML++ + TIML(ours) | 4-CONV | **52.19 ± 0.16%** | **69.02 ± 0.46%** | **51.97 ± 0.35%** | **71.06 ± 0.43%** |
| MAML + ALFABaik et al. (2020a) | 4-CONV | 50.58 ± 0.51% | 69.12 ± 0.47% | 53.16 ± 0.49% | 70.54 ± 0.46% |
| MAML + ALFA + TIML(ours) | 4-CONV | **51.07 ± 0.42%** | **69.67 ± 0.46%** | **54.77 ± 0.36%** | **72.04 ± 0.38%** |
| MAMLFinn et al. (2017b) | Resnet12 | 58.37 ± 0.49% | 69.76 ± 0.46% | 58.58 ± 0.49% | 71.24 ± 0.43% |
| MAML + TIML(ours) | Resnet12 | **59.52 ± 0.49%** | **72.85 ± 0.38%** | **60.32± 0.43%** | **75.72 ± 0.33%** |
| MAML + ALFABaik et al. (2020a) | Resnet12 | 59.74 ± 0.49% | 77.96 ± 0.41% | 64.63 ± 0.49% | 82.48 ± 0.38% |
| MAML + ALFA + TIML(ours) | Resnet12 | **59.55 ± 0.41%** | **78.62 ± 0.32%** | **65.42 ± 0.41%** | **84.59 ± 0.30%** |

### 4.3.2 CROSS-DOMAIN FEW-SHOT CLASSIFICATION

To further underscore the effectiveness of our TIML algorithm in adapting to unfamiliar tasks, we executed the following experiment as a cross-domain few-shot classification assessment. In this examination, both the meta-test tasks and meta-training tasks were carefully selected from dissimilar datasets exhibiting relatively low similarity. The results of this experiment have been summarized in Table 2. For this investigation, we adhered to the experimental protocol established by W.-Y. Chen et al.Chen et al. (2019) and employed CUB as the meta-test dataset to evaluate the adaptability of TIML's foundational model. This foundational model had undergone its initial training on miniImagenet, and we sought to evaluate its performance in the context of unknown tasks.

Table 2 displays the performance comparison among MAML, the recent MAML variant ALFA, and TIML, all of which were trained on the miniImageNet meta-training dataset and evaluated on the CUB meta-test dataset. Similar to the few-shot classification results presented in Table 1, TIML exhibits a significant enhancement in generalization performance, especially in the more challenging cross-domain few-shot classification scenario. Notably, TIML outperforms both MAML and ALFA + MAML by a greater margin in cross-domain few-shot classification than in few-shot classification. This highlights the effectiveness of TIML in learning new tasks from diverse domains and its robustness in bridging domain gaps, underscoring the importance of task-environment interactive training.

Another noteworthy observation from the results is that TIML demonstrates substantial improvements in generalization performance over both ALFA + MAML and MAML. This suggests that TIML addresses a distinct orthogonal problem. While ALFA [4] aims to enhance inner-loop optimization by focusing on developing new weight update rules (gradient descent), TIML concentrates on task-environment interactive training within inner-loop optimization. The consistent improvement in generalization performance across different baselines and architectures for TIML underscores the importance of well-designed task-environment interactive training and its robustness when combined with other models, showcasing its versatility and effectiveness.

### 4.3.3 ABLATION EXPERIMENT

In this section, we conducted an ablation study using the 4-CONV backbone in a 5-way 5-shot mini-classification scenario to better analyze the impact of the methods employed in TIML on the classification task outcomes.

To analyze the influence of different task selection methods within TIML, we performed experiments comparing four distinct approaches. The first approach involves averaging the training across all tasks without making any selection, adhering to the conventional meta-learning approach. The second approach involves random task selection without any specific criteria. The third approach

Table 2: Test accuracy on 5-way 5-shot cross-domain classification.

| Model | Baselearner | miniImagenet → CUB |
|---|---|---|
| | Backbone | 5-shot |
| MAMLFinn et al. (2017b) | 4-CONV | 52.70 ± 0.32% |
| MAML + TIML(ours) | 4-CONV | **54.18 ± 0.29%** |
| MAML + ALFABaik et al. (2020a) | 4-CONV | 58.35 ± 0.25% |
| MAML + ALFA + TIML(ours) | 4-CONV | **61.59 ± 0.19%** |
| MAMLFinn et al. (2017b) | Resnet12 | 53.83 ± 0.32% |
| MAML + TIML(ours) | Resnet12 | **56.68 ± 0.29%** |
| MAML + ALFABaik et al. (2020a) | Resnet12 | 61.22 ± 0.22% |
| MAML + ALFA + TIML(ours) | Resnet12 | **66.75 ± 0.21%** |

utilizes loss as the criterion for selection, selecting the task with the highest loss for training in each iteration. The fourth approach involves task-environment interactive learning, where task selection is made in real-time based on task environment information. This selection mechanism combines both greedy and random strategies.

The results of these four methods are presented in Table 3. It is evident that random task selection significantly reduces performance and is not a viable approach. The third and fourth methods yield nearly identical results, indicating no substantial difference between them, while both outperform the approach of training on all tasks uniformly. This underscores the effectiveness of our proposed task-environment interactive meta-learning. Task-environment interaction can be implemented in various ways, and in this study, we primarily used loss-based criteria to estimate future rewards. In the future, incorporating accuracy-based criteria may further enhance model performance.

Table 3: Experimental Accuracy of Different Choice Task Methods under 5-way5-shot.

| Different Selection Methods | Backbone | miniImagenet |
|---|---|---|
| Equal Training for All Tasks | 4-CONV | 63.11± 0.91% |
| Random Task Selection Training | 4-CONV | 52.13± 0.83% |
| Training on the Most Difficult Task | 4-CONV | 64.94± 0.35% |
| **Task-Environment Interactor(ours)** | 4-CONV | 65.10 ± 0.12% |

## 5 CONCLUSIONS

In this work, we introduce Task-Environment Interactive Meta-Learning (TIML), which can be applied to optimize gradient-based meta-learning frameworks. During training, we randomly sample tasks and select appropriate tasks for the current training iteration by considering the tasks' difficulty and expected future rewards. We also incorporate a random mechanism to mitigate overfitting. When integrated with various meta-learning algorithms, TIML consistently improves few-shot classification performance. The results from TIML underscore the critical importance of task-environment interactive training in few-shot learning. Without the need for extensive data preprocessing or significantly increasing the number of training iterations, selecting tasks that are beneficial for the current training iteration can lead to performance improvements.

In the natural world, events often occur randomly, and they are not altered by the subject's intention. Data preprocessing, which categorizes and pre-processes data before feeding it to a model, may not necessarily adhere to the principles of evolution. The results of task-environment interactive training demonstrate that task selection is more crucial than blindly increasing the training iterations. This approach exhibits strong generality and effectiveness. We believe that our findings can stimulate further interesting research. For instance, various discussions can be conducted regarding different task-environment interactive methods. Mechanisms for task-adaptive exploration of new tasks can also undergo many improvements.

## REPRODUCIBILITY STATEMENT

In order to promote and facilitate the reproducibility of our research findings, we have included the complete source code and supplementary materials as part of this paper. This comprehensive codebase, along with any additional resources, can be found in the supplementary materials. We believe that sharing our code and methodology not only ensures transparency in our research but also allows fellow researchers to replicate and build upon our work, ultimately contributing to the advancement of knowledge in our field.

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

## A    SELECTED EXPERIMENTAL RESULTS FIGURES

In the section on Algorithm Comparison, we present the experimental results of TIML in comparison to other algorithms. The left two columns showcase the advancements achieved by TIML on the 4-conv miniImagenet dataset. Meanwhile, the right two columns illustrate even more significant improvements observed with TIML on the Resnet12 tieredImageNet dataset. These results not only emphasize the effectiveness of TIML but also highlight its superior performance, particularly in challenging scenarios. Our findings underscore the potential of TIML as a promising algorithm in the field of machine learning and computer vision.

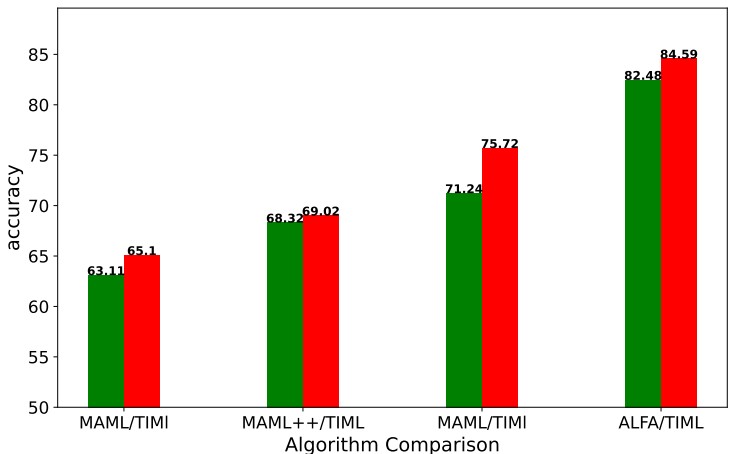

In our study, we conducted a series of ablation experiments to provide a comprehensive comparison of various task handling approaches. After randomly sampling tasks, we treat them differently, including random selection, averaging results after selecting all, choosing the most challenging task, and adopting TIML (a combination of random and greedy strategies). The results of these experiments unequivocally demonstrate the effectiveness of TIML in task handling and optimization. Notably, TIML outperforms other methods in our comparative analysis, underscoring its efficiency and versatility in addressing a wide range of tasks. These findings underscore TIML's significant contribution to improving task management and its potential impact on diverse applications.

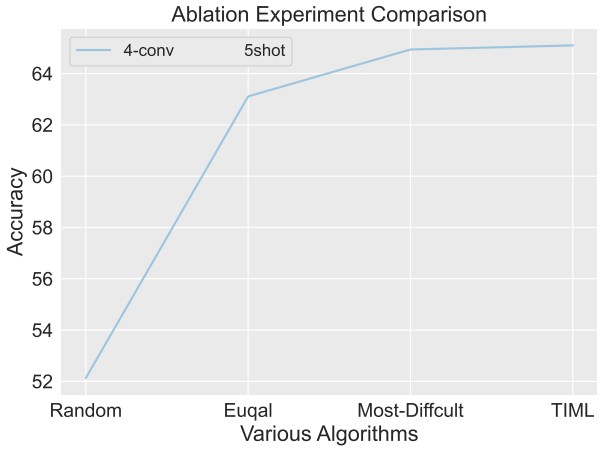

## B    DIFFERENCES FROM OTHER TASK SCHEDULING METHODS

In this section, we compare the differences and connections between six task scheduling methods and the interactive mechanisms of the task environment in this paper.

(1) The core idea of self-paced learning is to simulate human cognitive mechanisms by initially learning simple and universally applicable knowledge structures and gradually increasing difficulty, transitioning to more complex and specialized knowledge. However, the task interaction mechanism mentioned in this paper involves directly selecting more challenging tasks for learning, skipping the process of training on all tasks from easy to difficult. We adopt a selective training approach, balancing training resources among tasks with greater future returns while keeping the number of training iterations constant. The goal of meta-parameters is to establish a good starting point on different tasks. We begin with task assessments, prioritizing the training of tasks with higher difficulty and future returns. Information from tasks involved in the training is used to update meta-parameters, allowing them to converge more rapidly toward tasks with less favorable outcomes. This approach differs significantly from self-paced learning.

(2) The focal loss for dense object detection introduces a novel loss function, the focal loss, which is derived from modifying the standard cross-entropy loss. This function aims to focus the model's training on difficult-to-classify samples by reducing the weight of easily classifiable samples. The objective of this method aligns with the task interaction mechanism proposed in this paper, emphasizing concentration on more challenging situations. However, there are several differences in the specific approach. Firstly, focal loss operates on individual samples, while TIML operates on tasks. Secondly, focal loss employs a suitable function to measure the contribution of hard-to-classify and easy-to-classify samples to the overall loss, whereas TIML trains and evaluates tasks to update meta-parameters based on their difficulty and potential returns. Tasks that are relatively simple but yield better outcomes are essentially not selected and do not participate in the meta-parameter updating process. To avoid constantly selecting difficult tasks, we introduce a random selection mechanism, providing a chance for even simple tasks to be chosen. In summary, while there are some conceptual similarities between focal loss and TIML, their implementation methods and use cases are distinct.

(3) The method proposed in "Difficulty-aware Meta-Learning for Rare Disease Diagnosis (DAML)" is highly reminiscent of focal loss, as both involve adjusting weights for different situations. In meta-learning, task loss is used to update meta-parameters in reverse, whereas DAML introduces dynamically scaled cross-entropy loss for learning tasks. This automatically reduces the weight of simple tasks and focuses on challenging tasks. This aligns with the TIML concept presented in this paper, emphasizing a focus on difficult tasks, although the perspective on achieving this goal differs.

(4) In the three articles, "Adaptive Task Sampling for Meta-Learning", "Probabilistic Active Meta-Learning", and "Meta-Learning with an Adaptive Task Scheduler", as well as in the proposed Task Interaction Mechanism (TIML) in this paper, the methods share the perspective that tasks randomly sampled are suboptimal. The distinction lies in the fact that these three methods operate on a task pool containing all tasks, assuming that all tasks are known. They employ different approaches to assess and evaluate tasks before feeding them into the training process. In contrast, TIML, as proposed in this paper, makes superior choices for task training under the premise of random sampling, aiming to adapt to the environment without altering it. Meta-learning, belonging to the field of few-shot learning, faces challenges in acquiring samples, and the sample landscape continuously evolves over time. Continuous learning is required when new samples emerge. If operations are performed on the entire task pool, incorporating new samples requires mixing all samples for task sampling to achieve meaningful results, rendering previous training outcomes obsolete. With TIML, incorporating new samples into the previously trained results is sufficient for continued learning.

## C    SUMMARIZING THE CORE OF THE APPROACH IN THIS PAPER AND EXPLAINING THE CORE FORMULA

In the experimental section, a task is divided into two parts: a support set and a query set. The support set is used for training and updating task parameters, while the query set is used for evaluating the performance of task parameters. We use the task loss on the query set as the criterion, where a higher loss indicates poor training results. The goal of meta-learning is to have a good starting point for different tasks, meaning achieving good results with minimal gradient descent. We pass

the meta-parameters to different tasks within a batch, initially conducting a preliminary assessment for each task. The criterion for this evaluation is the task loss on the query set after one step of gradient descent on the task parameters for each task. A high task loss at this stage indicates that the meta-parameters are distant from optimal parameters for this task, suggesting a need for further in-depth training. We define a high loss as indicative of a challenging task. To avoid overfitting solely based on the current task difficulty, we introduce additional factors into the calculation of task difficulty. These factors include the future expectations from the previous step of the current task and the number of times the current task has been selected. This calculation yields the task's future reward. Based on this future reward, we choose a task for training. After training is complete, the process repeats, selecting a task for training based on its future reward in a loop.

During the training process, there may be situations where a particular task consistently performs poorly. According to the aforementioned approach, the system would keep training on this task, potentially getting stuck in a local minimum. To address this, we introduce a random selection mechanism to enhance generalization and avoid local traps.

What we truly need is a set of well-tuned meta-parameters. In this context, we have modified the meta-parameter updating process. We now exclusively utilize the query set losses from tasks that were selected for training during the task environment interaction mechanism to update the meta-parameters. Tasks that are selected for training indicate poor performance of the meta-parameters on these tasks. Therefore, we update the meta-parameters to be more aligned with these selected tasks.

In Formula 4, $\mathcal{L}_{\mathcal{T}_i}^{\mathcal{D}_i^Q}(t)$ represents the model's performance on the selected task in the query set. $R_i(t-1)$ denotes the cumulative reward for this task. A larger difference between these two values indicates a more significant improvement in training effectiveness. To ensure fair comparisons across different tasks, this difference is normalized by dividing it by the accumulated selection count for the chosen task. The updated future reward value is obtained by adding this normalized difference to the previous cumulative reward value.

## D  Performance Improvement of TIML in Complex Networks

The ResNet12 architecture has a more complex network structure compared to the 4-conv architecture, allowing it to learn richer and more abstract features for each task. Complex networks result in more precise outcomes during the training process, as shown in Table 1 of the experimental results, where the ResNet12 architecture demonstrates higher accuracy than the 4-conv architecture on the same dataset.

Similarly, as information for each task becomes more accurate, the details of conflicts between tasks become more pronounced. Our task environment interaction mechanism, which updates meta-parameters using information from the selected tasks during training, effectively reduces more task conflicts in the ResNet12 architecture compared to the 4-conv architecture, leading to a more noticeable improvement in performance.

Meta-parameters are designed to provide a good starting point for different tasks. Previous methods updated meta-parameters collectively for all tasks in a batch, reaching a balance point that couldn't favor distant tasks more rapidly. In contrast, our proposed method enables faster convergence of meta-parameters towards distant tasks, resulting in a better starting point. Similarly, in a more complex network, we not only mitigate more task conflicts but also leverage the finer details of features to rapidly enhance model performance.

