# OpenReview forum: "Meta-Learning with Task-Environment Interaction"
_ICLR.cc/2024/Conference — Submitted to ICLR 2024_

### Official Review · Reviewer_DThM · 2023-10-18

**Soundness:** 2 fair
**Presentation:** 3 good
**Contribution:** 1 poor
**Rating:** 3
**Confidence:** 3

**Summary:**

This paper studies the problem of harmful tasks in meta-model. The authors introduce a Task-Environment Interaction Meta-Learning(TIML) model, which feedback the task environment information to help to select the better task for updating the parameters. To avoid overfitting, they introduce a random parameter to randomly select a task for in-depth learning. Experiments are  conducted on the basis of MAML, and the result shows improvement.

**Strengths:**

There is a clear structure in the paper. The figures of the paper can present the structure and principles of the model.

**Weaknesses:**

First of all, there is no sufficient introduction of related work, which should include the recent research on meta-learning. And there is a typo in the title of Section 2 RELATED WORK.

Second, there is no elaboration of how the task information is passed into the meta-model to calculate the future utility.

Third, the authors only conducted experiments on the MAML model and its variant, which cannot show the improvement of TIML. There should be some experiments on the recent models to show the improvement of the TIML e.g. SKD, PAL.

**Questions:**

Can your model be combined with other models besides MAML? Would the experimental result improve in other model?

---

> ### Author Response · Authors · 2023-11-16
>
> The two methods proposed, SKD (Model-Agnostic Meta-Learning for Fast Adaptation of Deep Networks) and PAL (Partner-Assisted Learning for Few-Shot Image Classification), fall under the category of knowledge distillation approaches. They exhibit significant differences from the overall framework of MAML (Model-Agnostic Meta-Learning for Fast Adaptation of Deep Networks). However, all three methods belong to the realm of few-shot and meta-learning.
>     The approach presented in this paper can be applied to both SKD and PAL. In the field of meta-learning, the Task Interaction Mechanism (TIML) aims to make better decisions under the assumption of random task sampling. Knowledge distillation, on the other hand, involves creating a teacher model and a student model at the sample level, which contrasts with the random sampling approach proposed in this paper. The choice of random sampling is motivated by the need to adapt to the unpredictability of nature and foster learning in situations where the future environment cannot be foreseen.
>     The Task Interaction Mechanism proposed in this paper is applicable in scenarios with multiple parallel choices. SKD and PAL exhibit characteristics of mixing various tasks during task operations. Therefore, applying the proposed mechanism is likely to result in improved performance in these methods.
>    Thank you very much for your review. I will update the PDF in a few days, and I welcome your feedback and corrections.

---

> > ### Comment · Reviewer_DThM · 2023-11-23
> >
> > Thank the authors for the response. I appreciate your detailed explanation of your proposed method, which makes me have a clearer understanding of your thoughts. I would like to keep my initial score.

---

### Official Review · Reviewer_4N3x · 2023-10-18

**Soundness:** 2 fair
**Presentation:** 1 poor
**Contribution:** 2 fair
**Rating:** 3
**Confidence:** 2

**Summary:**

The authors propose a method for prioritizing certain tasks during meta-learning.

As far as I can tell, each time a batch of tasks is sampled, the method prioritizes tasks that seem to result in the highest intermediate losses during the course of the inner loop.

Various experiments show that this method improves the performance of MAML (and variants of it) somewhat.

**Strengths:**

- The method seems novel.

- The problem of task selection in meta-learning is important.

**Weaknesses:**

- The method does not do what it sets out to do. The authors claim that their method has an advantage over previous task-selection methods, in that it can handle the incoming tasks as they are provided, rather than constructing synthetic tasks as in other methods ("In the process of natural selection, we cannot choose whether tomorrow will be sunny or rainy;")

But that is not what is happening. The method requires that a batch of tasks can be sampled, and then that some of these tasks may be prioritized and others ignored during the meta-learning phase, which is equivalent to choosing whether you're seeing rain or sun, and basically similar to synthesizing tasks. Thus it is not clear what advantage is provided over existing task selection methods, including those mentioned by the authors themselves.

- The method is only compared to standard-MAML, and not to any alternative task selection method.

- The method is poorly described, making it difficult to understand (see below).

- Selecting tasks by difficulty is hardly a novel idea. Juergen Schmidhuber has long emphasized the need to concentrate on tasks that offer optimal difficulty (allowing for maximal learning, neither too hard nor too easy). More recently, the POET algorithm of Wang et al. (arXiv 1901.01753) largely selects new environments by their difficulty for the current generation of learners. (Note: I am not one of these authors)

This past research suggests that the method proposed by the authors (just favor the task with the highest intermediate loss) would fail badly on a very simple case: the presence of impossible or random tasks, which always generate a high loss without allowing for significant learning. This would largely eliminate learning under the method proposed here (but not standard MAML, which would still benefit from the non-impossible tasks).

- The actual algorithm, although confusingly reported, seems to be quite different from standard MAML. In particular, it seems that the gradient updates to the initialization parameters are computed and accumulated over every step of the inner loop ! Then the sum of these computed updates is applied to the initialization parameters in line 20 of Algo 1. This is quite different from MAML, which first goes through an entire inner loop (resulting in a single updated parameter theta_i), then computes a single loss over the "query set" (using theta_i), and takes the gradient of *that* over initialization parameters theta.


This makes the experiments uninformative about the benefit of the approach. If the standard MAML baselines were implemented by this same incremental method, they do not represent actual MAML performance. If, on the other hand, MAML was implemented in the standard way, we do not know whether the tiny improvements result from the task selection or from the novel (and puzzling) scheme of incremental outer-loop gradients.

- Minor: the paper is poorly written, with many typos. The algorithm is unclear (see below), and the discussion of related work is actually spread over sections 1, 2 and 3.

**Questions:**

- How do you think the algorithm would behave in the presence of impossible or random/uninformative tasks, as mentioned above?

- Confirm whether the outer-loop update (to the initialization parameters) is computed as the sum of intermediate gradients at each time step of the inner loop, which is what lines 14 and 20 in Algo 1 denote in their current form. How were the MAML baselines implemented?

- The text says that At selects the task with the highest *returns*. But the algorithm suggests that it is instead the task with the highest average *loss* (confirmed by the minus sign on the gradient). Which is it?

Minor:

- Section 2: "Related Word" -> Related Work.

- Many cases of citations where the parenthesis are at the wrong place (pro tip: redefine "\cite" as "\citet").

- Why switch from phi to theta in section 3.1?

- What are S and K in Algo 1?

- "Euqal" in the last figure of the appendix.

---

> ### Author Response · Authors · 2023-11-16
>
> Thank you very much for your review. During the training of a batch task group randomly sampled, only a few steps of gradient descent are employed. This process does not continuously consume a significant amount of resources for training. Additionally, to address the concern of random/uninformative tasks in a batch, the proposed task environment interaction mechanism incorporates a random selection process. This mechanism avoids persistent learning in tasks with poor performance.
>    Your characterization of the method presented in this paper as similar to those constructing synthetic tasks is not accurate. Currently, the synthesis of tasks that I am familiar with involves preprocessing the entire task pool, evaluating and judging tasks from various perspectives, and combining them to create new tasks. This approach incurs substantial resource costs. In contrast, the method proposed in this paper introduces a training decision during the training process without affecting the overall framework. Task selection is based on the premise that some tasks already demonstrate good performance on the query set, and thus, there is no need to train on them. The meta-learning discussed in this paper aims to find a starting point that is effective for the majority of tasks and quickly adapt it towards tasks with poorer performance.
>     The summing method used in line 20 of Algorithm 1 is consistent with MAML's Algorithm 1, line eight, both employing a summation. You may refer back to the original MAML paper for confirmation. The core innovation of the proposed task environment interaction mechanism lies in two aspects. Firstly, adjusting more optimal training strategies under the condition of random sampling. Secondly, altering the traditional outer loop update of MAML. Instead of using the sum of losses for all tasks in a batch to update the outer loop meta-parameters, we only use the summed losses of the selected training tasks. This accelerates the convergence towards tasks with poorer performance, enhancing overall generalization. Tasks that do not participate in the outer loop meta-parameter update are excluded because the meta-parameters already perform well on these tasks, and there is no need to bring them closer. We aim to find a starting point that is favorable for the majority of tasks.
>     I apologize if the wording has caused confusion. Some aspects of this paper may have been abrupt. The PDF will be updated in the next few days to address these issues. I hope this clarifies any confusion."

---

> > ### Comment · Reviewer_4N3x · 2023-11-17
> >
> > - The discrepancy is not line 20 of Algo 1, but line 14 - the computation of the gradients of the initialization weights over the query set at each step of the inner loop. No such line appears in the original MAML paper. Your presentation suggests that this is used for the computation of line 20 (since these gradients are apparently not used elsewhere in the algorithm?), which would of course be very wrong
> >
> > - The ATS approach mentioned by another reviewer involves selecting a sampling of tasks from a "candidate task pool" according to features of their final (post-inner loop) loss and gradients. Nothing suggests the candidate task pool should be the set of all tasks. In fact it needs not even be larger than batch size, if sampling is with replacement ! In the latter case the computational cost would be similar to that of your proposed method, proportional to Batch_size * S. It would, however, have the advantage of matching the actual MAML objective, unlike yours, as explained below.
> >
> > - Reading more carefully, it seems your algorithm will actually performs *multiple* successive inner loops, without reinitialization, on  any given task ! (Until its R ceases to be maximum, or $\epsilon$ probability triggers) If this is correct, your method essentially descends the meta-gradient of expected loss *after multiple inner loops*, specifically for hard tasks; which is quite different from MAML, which descends the meta-gradient of expected loss after a *single* inner loop over all tasks (few-shot learning). If this is correct, the departure from MAML is considerable and should be explicitly stated.
> >
> > - The problem of hard/impossible/uninformative task is hardly remedied by using an $\epsilon$, which I understand to be much lower than .5. Even if such tasks are rare, the system would still spend most of its time (and gradient steps) on them (unlike standard MAML, or even ATS due to the "sanity check" of query/support gradient similarity).
> >
> > - For these reasons, and also due to the lack of comparison with other task scheduling methods (as mentioned by other reviewers), I still believe that the paper is not appropriate for publication in its present form.

---

### Official Review · Reviewer_5KuL · 2023-10-27

**Soundness:** 2 fair
**Presentation:** 1 poor
**Contribution:** 1 poor
**Rating:** 3
**Confidence:** 5

**Summary:**

This study presents a meta-learning approach that employs an epsilon-greedy exploration strategy, actively choosing tasks by utilizing a reward function defined as the running average of each task's training loss. The efficacy of the proposed method is tested through its application to both standard and cross-domain few-shot classification tasks, where it exhibits enhancements over traditional Model-Agnostic Meta-Learning (MAML) techniques.

**Strengths:**

**Strength 1: Simplicity**

The proposed technique is simple, as it seamlessly integrates with existing meta-learning algorithms without necessitating any alterations to the foundational architecture. Notably, when applied to basic Model-Agnostic Meta-Learning (MAML) frameworks, this method consistently drives marginal performance enhancements.

**Weaknesses:**

**Weakness 1: Need for Improved Clarity and Precision**

The manuscript would benefit significantly from comprehensive revisions for enhanced clarity and precision in its presentation. It appears that the initial draft may have been produced with the assistance of a large language model, without post-editing.

One area requiring attention is the overall sentence structure and coherence within paragraphs. As it stands, some lengthy paragraphs, for example, the third paragraph of the introduction, are somewhat vague, hindering full comprehension. Additionally, many sentences lack proper spacing after periods.

Furthermore, the mathematical notations and equations present throughout the paper necessitate careful revision. There are instances of typographical errors, possibly arising from formatting issues in LaTeX or direct inputs from language models, which obscure the intended expressions and notations. This lack of precision makes it challenging for readers to grasp the concepts being discussed, particularly in Section 3.1, where most notations appear to be erroneously presented.

Specific examples of confusion include the inconsistent use of $\mathcal{T}\_i$ (collection of tasks or each task?) and the removal of brackets from $\\{ x_i^s, y_i^s\\}\_{s=1}^K$. The symbols $\varphi$ and $\theta$ seem to be used interchangeably, and there's inconsistent subscript formatting, as evidenced with subscript $i$. The notation following $S$ rounds of inner-loop updates also requires correction from $\theta\_{i,\delta}$ to the more appropriate $\theta\_{i,S}$.

Moreover, there is an interchangeable use of certain terms throughout the document, leading to potential confusion. This is observed in the inconsistent use of raw terms and their corresponding mathematical notations, such as $N$ vs. $\mathcal{N}$, $I$ vs. $\mathcal{I}$, $A$ vs. $\mathcal{A}$, and $R$ vs. $\mathcal{R}$. Additionally, the text could be clearer in its use of terms like $\mathcal{T}_i$ and $A_t$, as the current presentation may confuse readers.

In conclusion, the manuscript would substantially benefit from a detailed revision aimed at correcting these issues to improve the clarity and accuracy of both the textual and mathematical content. This process is essential for ensuring the work communicates its valuable insights more effectively to the readership.

**Weakness 2: Insufficient Justification and Elaboration of Central Concept**

A critical aspect of the manuscript that requires enhancement is the depth of rationale and exposition provided for the central idea, particularly concerning the reward function in Equation 4. While the concept as presented appears somewhat intuitive, the narrative lacks a detailed explanation or theoretical background that justifies this specific formulation.

**Weakness 3: Insufficient Novelty and Inadequate Comparative Analysis**
The paper significantly underperforms in establishing its novelty, particularly within the saturated domain of task scheduling algorithms, well-documented by sources [1-6]. A striking omission is the lack of a comprehensive comparison or substantive discussion related to these established works. Both the experimental and related work sections of the paper are noticeably thin on comparative analysis, undermining the paper's credibility and scholarly rigor.

In the experimental design, the authors restrict their focus to MAML-based methods, neglecting a host of other task scheduling approaches documented in [1-6]. This narrow scope fails to justify the proposed method's advantages within the broader context of the field. By not contrasting their results with these established methodologies, the authors miss the opportunity to demonstrate the superiority or the distinctive aspect of their approach.

At its core, the proposed method appears to be an oversimplified version of its predecessors, rather than a groundbreaking innovation. However, this perception may be somewhat influenced by the paper's lack of clarity, making it difficult to ascertain the full scope of the proposed methodology.

[1] SPL: Kumar et al., Self-paced learning for latent variable models, NeurIPS 2010.

[2] FOCAL: Lin et al., Focal loss for dense object detection, CVPR 2017.

[3] DAML: Li et al., Difficulty-aware meta-learning for rare disease diagnosis, MICCAI 2020.

[4] GCP: Liu et al., Adaptive task sampling for meta-learning, ECCV 2020.

[5] PAML: Kaddour et al., Probabilistic active meta-learning, NeurIPS 2020.

[6] ATS: Yao et al., Meta-learning with an adaptive task scheduler, NeurIPS 2021.

**Questions:**

Question 1: Could the authors provide a more detailed explanation of the rationale behind the reward formulation presented in Equation 4? Additionally, how does this formulation differentiate your approach from existing task scheduling strategies?

---

> ### Author Response · Authors · 2023-11-16
>
> In Formula 4, $\mathcal{L}_{\mathcal{T}_i}^{\mathcal{D}_i^Q}(t)$ represents the model's performance on the selected task in the query set. $R_i(t-1)$ denotes the cumulative reward for this task. A larger difference between these two values indicates a more significant improvement in training effectiveness. To ensure fair comparisons across different tasks, this difference is normalized by dividing it by the accumulated selection count for the chosen task. The updated future reward value is obtained by adding this normalized difference to the previous cumulative reward value.
>     The core idea of self-paced learning is to simulate human cognitive mechanisms by initially learning simple and universally applicable knowledge structures and gradually increasing difficulty, transitioning to more complex and specialized knowledge. However, the task interaction mechanism mentioned in this paper involves directly selecting more challenging tasks for learning, skipping the process of training on all tasks from easy to difficult. We adopt a selective training approach, balancing training resources among tasks with greater future returns while keeping the number of training iterations constant. The goal of meta-parameters is to establish a good starting point on different tasks. We begin with task assessments, prioritizing the training of tasks with higher difficulty and future returns. Information from tasks involved in the training is used to update meta-parameters, allowing them to converge more rapidly toward tasks with less favorable outcomes. This approach differs significantly from self-paced learning.
>    The focal loss for dense object detection introduces a novel loss function, the focal loss, which is derived from modifying the standard cross-entropy loss. This function aims to focus the model's training on difficult-to-classify samples by reducing the weight of easily classifiable samples. The objective of this method aligns with the task interaction mechanism proposed in this paper, emphasizing concentration on more challenging situations. However, there are several differences in the specific approach. Firstly, focal loss operates on individual samples, while TIMl operates on tasks. Secondly, focal loss employs a suitable function to measure the contribution of hard-to-classify and easy-to-classify samples to the overall loss, whereas TIMl trains and evaluates tasks to update meta-parameters based on their difficulty and potential returns. Tasks that are relatively simple but yield better outcomes are essentially not selected and do not participate in the meta-parameter updating process. To avoid constantly selecting difficult tasks, we introduce a random selection mechanism, providing a chance for even simple tasks to be chosen. In summary, while there are some conceptual similarities between focal loss and TIMl, their implementation methods and use cases are distinct.
>     The method proposed in "Difficulty-aware Meta-Learning for Rare Disease Diagnosis (DAML)" is highly reminiscent of focal loss, as both involve adjusting weights for different situations. In meta-learning, task loss is used to update meta-parameters in reverse, whereas DAML introduces dynamically scaled cross-entropy loss for learning tasks. This automatically reduces the weight of simple tasks and focuses on challenging tasks. This aligns with the TIML concept presented in this paper, emphasizing a focus on difficult tasks, although the perspective on achieving this goal differs.
>    In the three articles, "Adaptive Task Sampling for Meta-Learning," "Probabilistic Active Meta-Learning," and "Meta-Learning with an Adaptive Task Scheduler," as well as in the proposed Task Interaction Mechanism (TIML) in this paper, the methods share the perspective that tasks randomly sampled are suboptimal. The distinction lies in the fact that these three methods operate on a task pool containing all tasks, assuming that all tasks are known. They employ different approaches to assess and evaluate tasks before feeding them into the training process. In contrast, TIML, as proposed in this paper, makes superior choices for task training under the premise of random sampling, aiming to adapt to the environment without altering it. Meta-learning, belonging to the field of few-shot learning, faces challenges in acquiring samples, and the sample landscape continuously evolves over time. Continuous learning is required when new samples emerge. If operations are performed on the entire task pool, incorporating new samples requires mixing all samples for task sampling to achieve meaningful results, rendering previous training outcomes obsolete. With TIML, incorporating new samples into the previously trained results is sufficient for continued learning.
>   Thank you very much for your review. I will update the PDF in a few days, and I welcome your feedback and corrections.

---

> > ### Comment · Reviewer_5KuL · 2023-11-21
> >
> > Thank you for your detailed response to my initial review. I appreciate the effort put into addressing the concerns raised. However, I found it challenging to follow the rebuttal due to its length and lack of clear organization. For future submissions, I recommend structuring the response more systematically, perhaps by directly aligning your points with the specific concerns raised in the review.
> >
> > Regarding the TIML methodology, while I now have a better understanding of how it diverges from previous methods, I still find the justification for its preference over other methods unclear. This is particularly due to the absence of intuitive explanations and empirical comparisons.
> >
> > Lastly, my primary concern about the significant imprecision in definitions and notations of key terms remains unaddressed, as the current manuscript does not reflect any updates. This aspect is crucial for the clarity and rigidity of the research. Unfortunately, due to these unresolved issues, I maintain my initial rating.

---

### Official Review · Reviewer_E2rE · 2023-10-28

**Soundness:** 3 good
**Presentation:** 3 good
**Contribution:** 3 good
**Rating:** 6
**Confidence:** 3

**Summary:**

This paper introduces Task-Environment Interactive Meta-Learning (TIML) as a way to optimise gradient-based meta-learning frameworks. During training, TIML selects tasks based on their difficulty and expected future rewards, incorporating a random mechanism to prevent overfitting.

**Strengths:**

The paper's originality lies in its novel approach to meta-learning, particularly the Task-Environment Interactive Meta-Learning (TIML) method. It introduces a fresh perspective by addressing the challenge of task selection during meta-training, which is an innovative way to enhance few-shot learning.

**Weaknesses:**

The paper mentions that task selection is based on factors like task difficulty and expected future rewards, but it doesn't provide a detailed discussion or analysis of how these criteria are determined or how they impact the selection process. A more comprehensive exploration of the decision-making process for task selection, along with an analysis of the sensitivity of TIML's performance to variations in these criteria, would enhance the understanding of the method's inner workings.

**Questions:**

In the context of the TIML algorithm's experiment on cross-domain few-shot classification, why is it significant that TIML exhibits more substantial improvements in the ResNet12 architecture compared to the 4-CONV architecture, and what does this observation suggest about the algorithm's performance in more complex network structures?

---

> ### Author Response · Authors · 2023-11-16
>
> In the experimental section, a task is divided into two parts: a support set and a query set. The support set is used for training and updating task parameters, while the query set is used for evaluating the performance of task parameters. We use the task loss on the query set as the criterion, where a higher loss indicates poor training results. The goal of meta-learning is to have a good starting point for different tasks, meaning achieving good results with minimal gradient descent. We pass the meta-parameters to different tasks within a batch, initially conducting a preliminary assessment for each task. The criterion for this evaluation is the task loss on the query set after one step of gradient descent on the task parameters for each task. A high task loss at this stage indicates that the meta-parameters are distant from optimal parameters for this task, suggesting a need for further in-depth training. We define a high loss as indicative of a challenging task. To avoid overfitting solely based on the current task difficulty, we introduce additional factors into the calculation of task difficulty. These factors include the future expectations from the previous step of the current task and the number of times the current task has been selected. This calculation yields the task's future reward. Based on this future reward, we choose a task for training. After training is complete, the process repeats, selecting a task for training based on its future reward in a loop.
>     During the training process, there may be situations where a particular task consistently performs poorly. According to the aforementioned approach, the system would keep training on this task, potentially getting stuck in a local minimum. To address this, we introduce a random selection mechanism to enhance generalization and avoid local traps.
> What we truly need is a set of well-tuned meta-parameters. In this context, we have modified the meta-parameter updating process. We now exclusively utilize the query set losses from tasks that were selected for training during the task environment interaction mechanism to update the meta-parameters. Tasks that are selected for training indicate poor performance of the meta-parameters on these tasks. Therefore, we update the meta-parameters to be more aligned with these selected tasks.
>    The ResNet12 architecture has a more complex network structure compared to the 4-conv architecture, allowing it to learn richer and more abstract features for each task. The complex network yields more precise results during the training process, as evidenced by higher accuracy on the same dataset in ResNet12 compared to the 4-conv architecture. However, with increased complexity comes more detailed conflict information between tasks. Our task environment interaction mechanism, which utilizes information only from tasks selected for training during the process, effectively reduces task conflicts more in ResNet12 than in the 4-conv architecture, resulting in a more pronounced performance improvement.
>     Meta-parameters are designed to provide a good starting point for different tasks. Previous methods updated meta-parameters collectively for all tasks in a batch, reaching a balance point that couldn't favor distant tasks more rapidly. In contrast, our proposed method enables faster convergence of meta-parameters towards distant tasks, resulting in a better starting point. Similarly, in a more complex network, we not only mitigate more task conflicts but also leverage the finer details of features to rapidly enhance model performance.
>    Thank you very much for your review. I will update the PDF in a few days, and I welcome your feedback and corrections.

---

### Meta-Review · Area_Chair_ByZz · 2023-12-05

**Metareview:**

This paper presents a meta-learning method where the task selection is done actively in the meta-training phase. The technique in this paper can be easily integrated with existing meta-learning methods, which is good. However, there are a few critical concerns raised by reviewers. Of particular concerns are unclarities in the writing, insufficient justification of concept, inadequate comparative analysis. The authors made efforts in providing the rebuttal, so that the reviewers feel that they better understand the paper. Some of critical concerns were not properly addressed, leading that all reviewers kept their initial scores. We also feel that it is quite challenging to fully revise the paper in the limited amount of time, accommodating all the comments. We feel that the paper is not ready for being published on ICLR in its current version. Therefore, the paper is not recommended for acceptance in its current form. I hope authors found the review comments informative and can improve their paper by addressing these carefully in future submissions.

**Justification For Why Not Higher Score:**

needs clear justification and much improved presentation.

**Justification For Why Not Lower Score:**

N/A

---

### Decision · Program_Chairs · 2024-01-16

Reject